# Single-cell transcriptional dynamics of flavivirus infection

**Fabio Zanini[1†]\*, Szu-Yuan Pu[2,3†], Elena Bekerman[2,3], Shirit Einav[2,3‡], Stephen R Quake[1,4,5‡]\***

[1]Department of Bioengineering, Stanford University, Stanford, United States; [2]Division of Infectious Diseases, Department of Medicine, Stanford University School of Medicine, Stanford, United States; [3]Department of Microbiology and Immunology, Stanford University School of Medicine, Stanford, United States; [4]Department of Applied Physics, Stanford University, Stanford, United States; [5]Chan Zuckerberg Biohub, San Francisco, United States

**Abstract** Dengue and Zika viral infections affect millions of people annually and can be complicated by hemorrhage and shock or neurological manifestations, respectively. However, a thorough understanding of the host response to these viruses is lacking, partly because conventional approaches ignore heterogeneity in virus abundance across cells. We present viscRNA-Seq (virus-inclusive single cell RNA-Seq), an approach to probe the host transcriptome together with intracellular viral RNA at the single cell level. We applied viscRNA-Seq to monitor dengue and Zika virus infection in cultured cells and discovered extreme heterogeneity in virus abundance. We exploited this variation to identify host factors that show complex dynamics and a high degree of specificity for either virus, including proteins involved in the endoplasmic reticulum translocon, signal peptide processing, and membrane trafficking. We validated the viscRNA-Seq hits and discovered novel proviral and antiviral factors. viscRNA-Seq is a powerful approach to assess the genome-wide virus-host dynamics at single cell level.
DOI: https://doi.org/10.7554/eLife.32942.001

**\*For correspondence:**
fabio.zanini@stanford.edu (FZ);
quake@stanford.edu (SRQ)

[†]These authors contributed equally to this work
[‡]These authors also contributed equally to this work

**Competing interests:** The authors declare that no competing interests exist.

## Introduction

*Flaviviruses*, which include dengue (DENV) and Zika (ZIKV) viruses, infect several hundred million people annually and are associated with severe morbidity and mortality (*Bhatt et al., 2013*; *Rasmussen et al., 2016*; *Guzman and Kouri, 2003*). Attempts to develop antiviral drugs that target viral proteins have been hampered in part by the high genetic diversity of flaviviruses. Since viruses usurp the cellular machinery at every stage of their life cycle, a therapeutic strategy is to target host factors essential for viral replication (*Bekerman and Einav, 2015*). To this end it is paramount to understand the interaction dynamics between viruses and the host, to identify pro- and antiviral host factors and to monitor their dynamics in the course of viral infection. The current model of flavivirus infection suggests that the virus enters its target cells via clathrin-mediated endocytosis, followed by RNA genome uncoating in the early endosomes and trafficking to ER-derived membranes for translation and viral RNA replication. Following assembly, viral particles are thought to bud into the ER lumen and are then released from the cell via the secretory pathway (*Screaton et al., 2015*). This pattern notwithstanding, it remains a challenge to determine the entire complement of host genes that interact, either directly or indirectly, with DENV or ZIKV.

Several high-throughput approaches have been applied to screen all 20,000 human genes for interactions with flaviviruses, including knockdown screens based on RNA interference (*Sessions et al., 2009*; *Kwon et al., 2014*; *Le Sommer et al., 2012*), knockout screens via haploid cell lines or CRISPR libraries (*Marceau et al., 2016*; *Zhang et al., 2016*; *Lin et al., 2017*), and bulk

transcriptomics via microarrays or RNA-Seq (*Sessions et al., 2013*; *Moreno-Altamirano et al., 2004*; *Fink et al., 2007*; *Conceição et al., 2010*; *Becerra et al., 2009*; *Liew and Chow, 2006*). While these approaches have provided important insights, our understanding of infection-triggered cellular responses is far from complete.

Knockdown, knockout, and population-level transcriptomics screens are extremely valuable tools but also share some limitations. First, because they are bulk assays, the heterogeneity of virus infection in single cells is obscured in the averaging process; differences in timing of virus entry and cell state across the culture and the fraction of uninfected cells are not accounted for. Second, because each population is a single data point and experiments cannot be repeated more than a handful of times, reproducibility and batch effects represent a challenge. Third, in knockout and knockdown screens the temporal aspect of infection is largely ignored, because successful knockdown can take days and recovery of the culture after infection in knockout screens lasts even longer. Fourth, because both knockdown and knockout can impair cellular viability and cannot probe essential genes, only a subset of genes can be probed by these techniques.

Here we report the development of viscRNA-Seq, an approach to sequence and quantify the whole transcriptome of single cells together with the viral RNA (vRNA) from the same cell. We applied this platform to DENV and ZIKV infections and investigated virus-host interactions in an unbiased, high-throughput manner, keeping information on cell-to-cell variability (i.e. cell state) and creating statistical power by the large number of single cell replicates while avoiding essential gene restrictions. By correlating gene expression with virus level in the same cell, we identified several cellular functions involved in *flavivirus* replication, including ER translocation, N-linked glycosylation and intracellular membrane trafficking. By comparing transcriptional dynamics in DENV versus ZIKV infected cells, we observed great differences in the specificity of these cellular factors for either virus, with a few genes including ID2 and HSPA5 playing opposite roles in the two infections. Using loss-of-function and gain-of-function screens we identified novel proviral (such as RPL31, TRAM1, and TMED2) and antiviral (ID2, CTNNB1) factors that are involved in mediating DENV infection. In summary, viscRNA-Seq sheds light on the temporal dynamics of virus-host interactions at the single cell level and represents an attractive platform for discovery of novel candidate targets for host-targeted antiviral strategies.

## Results

### viscRNA-Seq recovers mRNA and viral RNA from single cells

viscRNA-Seq is modified from the commonly used Smart-seq2 for single cell RNA-Seq (*Picelli et al., 2014*). Briefly, single human cells are sorted into 384-well plates pre-filled with lysis buffer (*Figure 1C*). In addition to ERCC (External RNA Controls Consortium) spike-in RNAs and the standard poly-T oligonucleotide (oligo-dT) that captures the host mRNA, the lysis buffer contains a DNA oligo that is reverse complementary to the positive-strand viral RNA (*Figure 1D*). The addition of a virus-specific oligo overcomes limitations of other approaches and enables studying of viruses that are not polyadenylated (*Russell et al., 2018*). Reverse transcription and template switching is then performed as in Smart-seq2, but with a 5'-blocked template-switching oligonucleotide (TSO) that greatly reduces the formation of artifact products (TSO concatemers). The cDNA is then amplified, quantified, and screened for virus presence via a qPCR assay (*Figure 1E*). Since many cells are not infected, this enables us to choose wells that contain both low and high vRNA levels and then to sequence their cDNA on an illumina NextSeq at a depth of ~ 400,000 reads per cell (*Figure 1F*). This approach provides high coverage of transcriptome and allows high-quality quantitation of gene expression and intracellular virus abundance in a relatively large number of cells.

We applied viscRNA-Seq to an infection time course in cultured cells. We infected human hepatoma (Huh7) cells with DENV (serotype 2, strain 16681) at multiplicity of infection (MOI) of 1 and 10. To assess reproducibility, we performed an independent experiment on DENV infection on a smaller scale (1/5th of the cell numbers) and obtained consistent results (see *Figure 2—figure supplement 1*). In a separate experiment, Huh7 cells were infected with ZIKV (Puerto Rico strain, PRVABC59) at an MOI of 1. Uninfected cells from the same culture were used as controls (*Figure 1A*). At four different time points after infection – 4, 12, 24, and 48 hr – cells were harvested, sorted, and processed with viscRNA-Seq (*Figure 1B*). Recovery of the ERCC spike-ins and number of expressed genes per

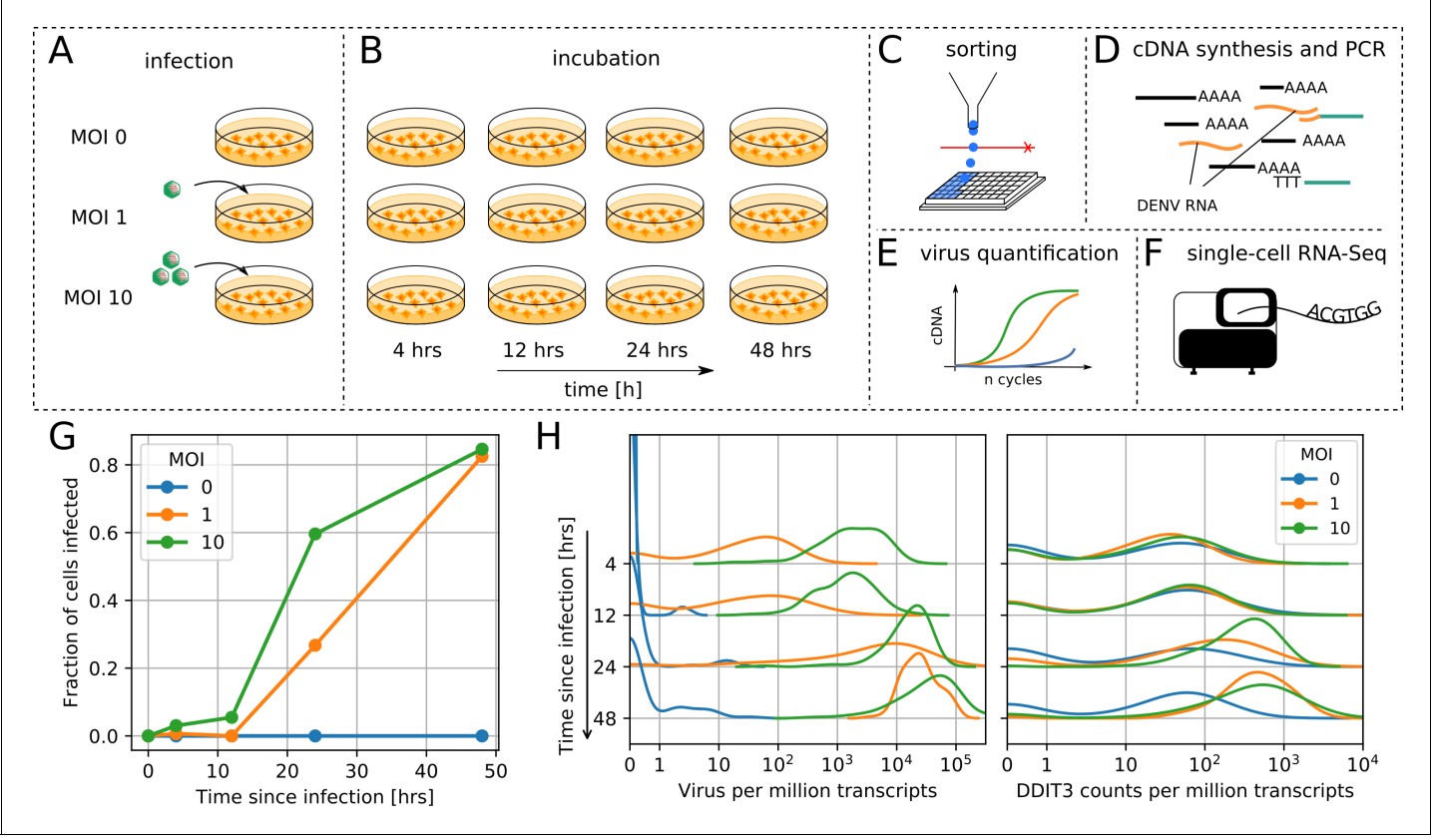

**Figure 1.** viscRNA-Seq quantifies gene expression and virus RNA from the same cell. . (A to F) Experimental design: (**A**) human hepatoma (Huh7) cells are infected with dengue or Zika virus at time 0 at multiplicity of infection (MOI) 0 (control), 1, or 10, then (**B**) harvested at different time points, (**C**) sorted and lysed into single wells. (**D**) Both mRNA and viral RNA (vRNA) are reverse transcribed and amplified from each cell, then (**E**) cells are screened for virus infection by qPCR. (**F**) Libraries are made and sequenced on an illumina NextSeq 500 with a coverage of ~400,000 reads per cell. (**G**) The fraction of cells with more than 10 virus reads increases with MOI and time, saturating at 48 hr post infection. (**H**) Distributions of number of virus reads (left) and expression of an example stress response gene (right) inside single cells, showing the different dynamics of pathogen replication and host response. Whereas virus content can increase 1000 fold and shows no saturation, expression of DDIT3/CHOP saturates after a 10 fold increase.

DOI: https://doi.org/10.7554/eLife.32942.002

The following figure supplement is available for figure 1:

**Figure supplement 1.** Quality Controls of the viscRNA-Seq approach.

DOI: https://doi.org/10.7554/eLife.32942.003

cell confirmed that the libraries were of high quality (*Figure 1—figure supplement 1*, panels A-B). From each experimental condition, 380 cells were screened for virus and ~100 of those were sequenced. In total ~7500 single cells were screened and ~2100 were sequenced (see *Supplementary file 1*).

## Intracellular virus abundance and gene expression are heterogeneous across cells

First, we focused on infection by DENV. As expected, qPCR showed an increase in the fraction of infected cells with both MOIs over time (*Figure 1G*). Whereas most genes were rather homogeneously expressed, both intracellular virus abundance (number of vRNA reads per million transcripts) and expression of a subset of genes varied widely across infected cells (*Figure 1H*). Overall, between zero and a quarter of all reads from each cell (i.e. ~$10^5$ reads) are vRNA-derived, hence the dynamic range for intracellular virus abundance is extremely wide. On average, intracellular virus amount increased with time and MOI. The distribution of both intracellular virus abundance and gene expression are rather symmetric in logarithmic space (*Figure 1H*); as a consequence, mean expression as measured in a bulk assay is higher than the median and over-represents highly

infected cells. The high coverage sequencing enables a quantitative measurement of the variation in the expression level of thousands of genes in each cell (*Figure 1—figure supplement 1–1B*). As a next step, we aimed at identifying which elements of this variation are induced by the infection.

## Correlation between intracellular virus abundance and gene expression within single cells tracks infection-triggered host response

In a bulk assay each of the experimental conditions would be an average of all cells, making it difficult to extract clear statistical patterns. Leveraging both single-cell resolution and high throughput, we directly computed Spearman's rank correlation coefficient between each gene expression and intracellular virus abundance across all cells. This metric does not require an explicit noise model for either expression or virus abundance and is therefore insensitive to outlier cells. To assess uncertainties, we performed 100 bootstraps over cells (see Materials and methods). As expected, most genes do not correlate with vRNA level and the distribution of their correlation coefficients decays rapidly away from zero (*Figure 2A*). In panels 2B-D examples of strong anticorrelation, strong correlation, and absence of correlation are shown. Both the level of vRNA at which each gene starts to correlate and the slope of the response vary across genes and may reflect different infection stages (see below). Genes with extreme correlation consistently represent specific cellular functions. Most of the top correlated genes (*Figure 2A* right inset) are involved in the ER unfolded protein response (UPR) (see e.g. DDIT3 in *Figure 2C*), consistent with ER stress response triggered by flavivirus translation and RNA replication on ER-derived membranes (*Medigeshi et al., 2007*). Numerous strongly anticorrelated genes (*Figure 2A* left inset) are components of actin and microtubules, indicating cytoskeleton breakdown (as an example, see ACTB in *Figure 2B*). Notice that anticorrelated genes appear to react at higher intracellular virus amounts than correlated genes, as exemplified by the higher threshold for ACTB than DDIT3 (see *Figure 2B–C*, Materials and methods, and *Figure 2—figure supplements 2,3*). Molecular chaperones are found in both categories suggesting a more nuanced regulation.

To understand whether correlated genes may represent pathways that are important for virus infection, we focused on the top 1% correlated subset of the transcriptome (correlation in excess of 0.3 in absolute value) and performed Gene Ontology (GO) enrichment analysis using the online service PANTHER (*Mi et al., 2017*). This statistical analysis confirmed the qualitative picture emerging from the top correlates. At 4 hr post-infection upregulation of genes involved in translation and suppression of mRNA processing is demonstrated. At 48 hr post-infection there is an upregulation of UPR, protein degradation via ERAD, and ER-to-Golgi anterograde transport via COPII-coated vesicles, and a downregulation of cytoskeleton organization and cell cycle genes related to both G1-S and G2-M phases (see *Supplementary files 1–4*). No clear effect of cell cycle genes on infection at early time points is observed, in agreement with previous reports in human cells (see *Figure 2—figure supplement 4*) (*Helt and Harris, 2005*).

## Several genes switch role during dengue infection

Naturally, cells that are infected for longer tend to harbor more vRNA. To disentangle the effect of time since infection from the vRNA level within each cell, we computed the same correlation coefficient within single time points. We discovered that most correlated genes exhibit either positive or negative correlation, but not both. This behavior is expected for generic stress response genes; the sign of the differential expression is a hardwired component of their physiological function. However, a group of 17 'time-switcher' genes show both an anticorrelation of less than −0.3 and a correlation in excess of +0.3 at different time points post-infection, suggesting a more specific interaction with DENV. Of these, six genes transition from anticorrelation to correlation (e.g. COPE, *Figure 2E–F*), 10 show the opposite trend, and a single gene (PFN1) follows a nonmonotonic pattern (*Figure 2G*). Since more than two time points were sampled, a consistent increase (or decrease) in correlation likely stems from a biological change rather than a technical noise. Of the six proteins which switch from anticorrelated to correlated, RPN1 and HM13 localize to the ER. RPN1 is a noncatalytic member of the oligosaccharide transfer (OST) complex, which is required for N-linked glycosylation of some ER proteins, whereas HM13 is a protease that cleaves the signal peptide after translocation into the ER. Both of these factors have been shown to be essential for DENV infection (*Marceau et al., 2016*). Of the other four proteins that show a similar behavior, SQSTM1 is a scaffold

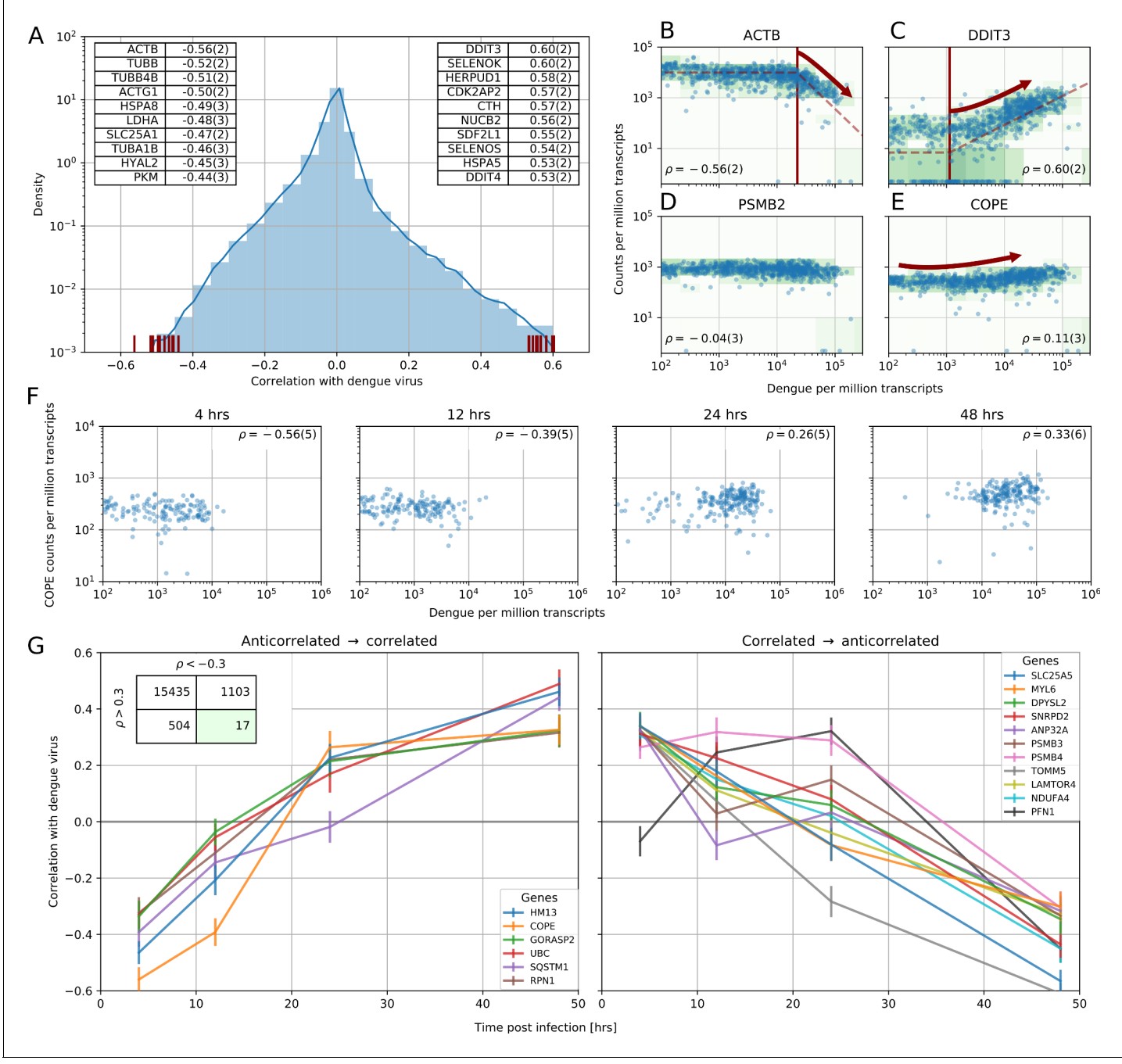

**Figure 2.** Correlation between dengue vRNA and gene expression reveals cellular processes involved in dengue virus infection. (**A**) Distribution of Spearman correlation coefficients between dengue vRNA and mRNA from the same cell across all human genes. The insets list the top correlated (right) and anticorrelated (left) genes. Response to ER stress and apoptosis is activated as infection proceeds, whereas actin and microtubules pathways are downregulated. (**B–E**) Examples of correlation patterns observed across the transcriptome, as a scatter plot of vRNA amount versus gene expression. Each dot is a single cell and the green shades indicate the density of cells. Dashed lines indicate least-square piecewise-linear fits in log-log space (see Materials and methods): (**B**) Anticorrelation at high vRNA content, (**C**) correlation at medium to high vRNA content, (**D**) no correlation, and (**E**) time-dependent correlation dynamics. (**F**) Expression versus vRNA content for gene COPE, as shown in panel E but splitting cells by time after infection. Correlation at each time is shown in the top left corner of each plot, and switches from strongly negative to strongly positive as infection proceeds. (**G**) Correlation between expression and dengue vRNA content switches from negative to positive (< −0.3 to >+0.3) for six genes (left panel) and in the opposite direction for 11 genes (right panel), highlighting potential multiple roles of these genes during dengue virus infection. Error bars and numbers in parentheses are standard deviations of 100 bootstraps over cells (the latter indicates uncertainties on the last digit).

DOI: https://doi.org/10.7554/eLife.32942.004

*Figure 2 continued on next page*

*Figure 2 continued*

The following figure supplements are available for figure 2:

**Figure supplement 1.** An independent smaller scale, time course DENV infection experiment shows consistent results across replicates in terms of genes that are correlated and anticorrelated with intracellulal virus abundance.
DOI: https://doi.org/10.7554/eLife.32942.005

**Figure supplement 2.** Parametric fitting of piecewise-linear gene expression versus intracellular virus amount infers reaction thresholds for infected cells.
DOI: https://doi.org/10.7554/eLife.32942.006

**Figure supplement 3.** Genes that are anticorrelated with intracellular virus amount change expression at a higher threshold than genes that are positive correlated.
DOI: https://doi.org/10.7554/eLife.32942.007

**Figure supplement 4.** Cell cycle phase does not appear to affect intracellular DENV abundance.
DOI: https://doi.org/10.7554/eLife.32942.008

**Figure supplement 5.** Bystander effects are not significant but suggestive for the COPE gene.
DOI: https://doi.org/10.7554/eLife.32942.009

**Figure supplement 6.** Gene expression versus vRNA level across all time points and MOIs during DENV infection for members of the translocon (SEC61), TRAP complex, signal recognition particle (SRP), signal peptidase complex (SPCS), oligosaccharide transfer complex (OST), plus two ribosomal proteins and two more proteins involved in ER translocation (HM13 and TRAM1).
DOI: https://doi.org/10.7554/eLife.32942.010

protein involved in selective autophagy of polyubiquitinated substrates and has been shown to have a bimodal behavior in DENV infection (*Metz et al., 2015*), whereas UBC is a major source of ubiquitin. Lastly, GORASP2 and COPE play a role in Golgi assembly and/or membrane trafficking. In particular COPE is a subunit of the coatomer complex (COPI) that mediates both intra-Golgi and Golgi-to-ER retrograde vesicle transport; another subunit of this complex, COPB1, has recently been shown to be essential for DENV (*Iglesias et al., 2015*). Interestingly, COPE also appears to be downregulated during early infection in 'bystander' cells; i.e. cells that originate from an infected culture but are themselves not infected (*Figure 2—figure supplement 5*). No uninfected cells were recovered from infected cultures at late time points.

## Host response difference between DENV and ZIKV infections

Next we sought to address the question of which elements of the host response are common between DENV and ZIKV, and therefore potentially common with other evolutionarily related viruses as well. To do so, we replicated the time course experiment with ZIKV at MOI 0 (control) and 1. Although Huh7 cells were also infected at an MOI of 10, cell death precluded sorting. *Figure 3A* shows the correlations between gene expression (each dot represents a human gene) and vRNA for both experiments and represents the two-dimensional equivalent of *Figure 2A*. We discovered that the majority of genes are not correlated with either virus (contour lines indicate density of genes). Nevertheless, a clear pattern with genes along the positive diagonal emerged, such as ATF3 (*Figure 3C*) and ACTG1 (*Figure 3D*), demonstrating a similar behavior upon infection with either virus. A minority of genes are scattered away from the diagonal, indicating discordant behavior between DENV or ZIKV infection. For instance, ID2 expression decreases at high DENV level but increases at high ZIKV RNA level (*Figure 3B*), while the opposite trend is observed with the chaperone HSPA5 (*Figure 3E* and see below). A number of genes at the outskirts of the correlation plot are labeled and highlighted in red as they exhibit noteworthy expression patterns upon infection: i.e. either an extremely strong correlation with both viruses or a high degree of virus specificity. These outliers include two subunits of the SEC61 complex (B, and G), several subunits of the TRAP complex and the OST, previously shown to be essential for DENV and/or WNV infection (*Marceau et al., 2016*; *Zhang et al., 2016*), and other genes that may be relevant to infection with either virus.

To understand how these correlated genes shape the heterogeneity of infected cells, we selected all genes with a correlation coefficient above 0.4 or below −0.4 and performed t-SNE dimensionality reduction (*Maaten and Hinton, 2008*), coloring each cell by its intracellular virus abundance (*Figure 4A*, left) or by time post-infection (right). Although uninfected cells form a mixed, heterogeneous cloud, infection pushes cells into more stereotypic states that are distinct for DENV and ZIKV

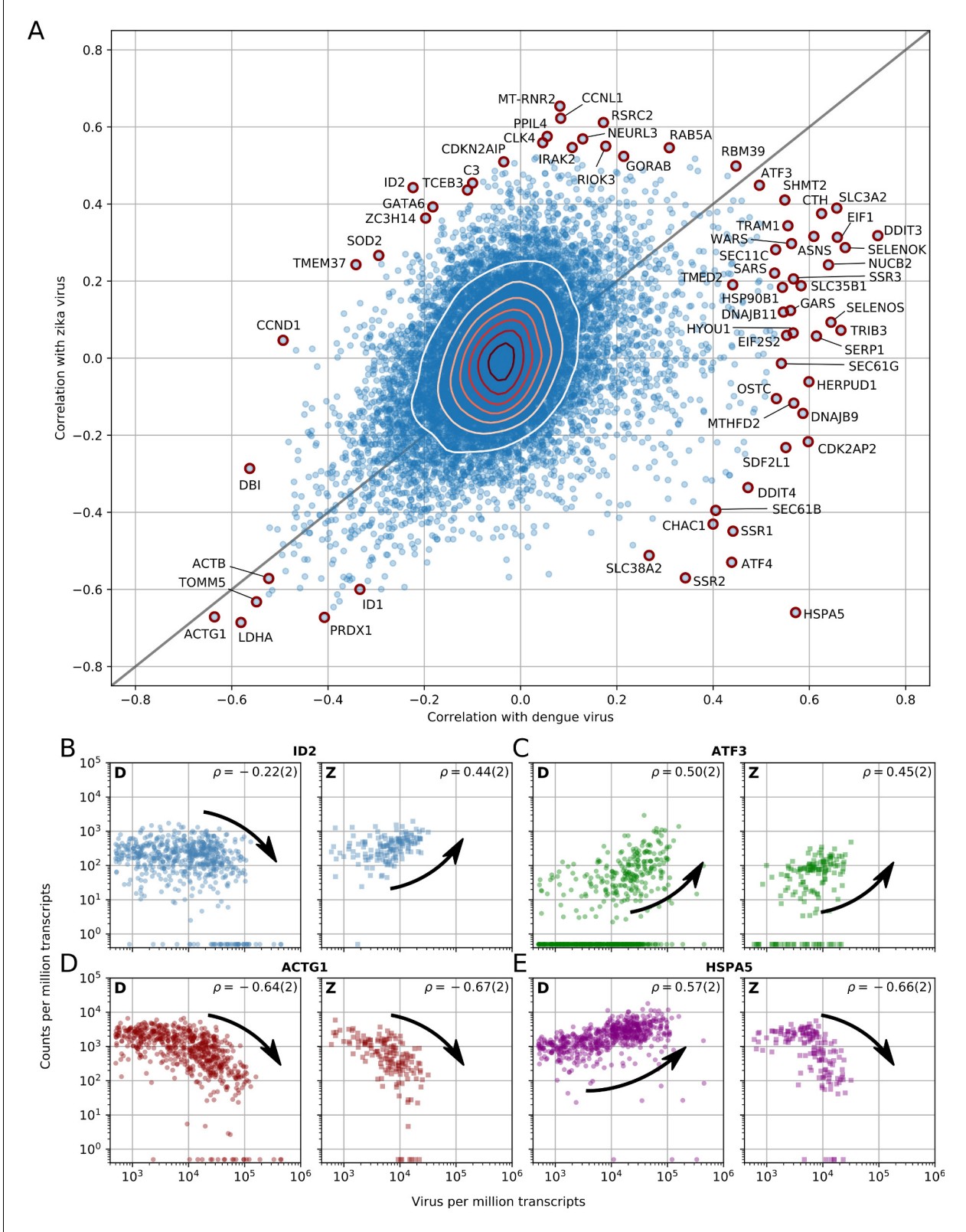

**Figure 3.** Dengue and Zika virus induce partially overlapping cellular responses. (**A**) Correlation between gene expression and vRNA during Dengue virus versus Zika virus infection. Each dot is a gene and the contour lines indicate the an estimate of the density of genes. Most genes do not correlate with either virus, but some genes correlate strongly with different degrees of virus specificities. Only cells with 500 or more virus reads per million transcripts are used for this analysis (see main text). (**B–E**) Examples of genes with different behavior across the two viruses, as a scatter plot of gene

*Figure 3 continued on next page*

*Figure 3 continued*

expression versus vRNA content. Each dot is a single cell. Dengue plots are indicated by a D, Zika plots by a Z in the top left corner. Numbers in parentheses are standard deviations of 100 bootstraps over cells (uncertainties on the last digit).

DOI: https://doi.org/10.7554/eLife.32942.011

The following figure supplements are available for figure 3:

**Figure supplement 1.** Cells with the highest intracellular DENV amounts show clear signs of prolonged ER stress response but no obvious sign of increased apoptosis.

DOI: https://doi.org/10.7554/eLife.32942.012

**Figure supplement 2.** Cells with the highest intracellular ZIKV amounts show ER stress response, increased CASP3 and reduced ATF4 expression.

DOI: https://doi.org/10.7554/eLife.32942.013

infection (black arrows indicate average positions for cells at increasing intracellular virus abundance). t-SNE visualization represents global trends contributed by many genes; plotting gene expression dynamics on top of these visualizations enables one to connect single genes to these widespread changes defined by virus infection (*Figure 4B*). Remarkably, the temporal behavior of a few genes is inconsistent with the global transcriptomics shifts: for instance the expression of HSPA5 increases until 24 hr after ZIKV infection, but is then sharply decreased at 48 hr post-infection and with higher intracellular virus abundance. To compare the temporal dynamics of gene expression during DENV and ZIKV infection, we identified 'time switchers' for Zika infection (*Figure 4C*). Although the number of genes that show both correlation and anticorrelation is similar between the two viruses, the 11 Zika time switchers exhibit no correlation at 4 hr post-infection, followed by a non-monotonic behavior as time passes. HSPA5 is included in this list, in agreement with its t-SNE visualization; this gene is therefore not only subject to opposite regulation in DENV versus ZIKV infection, but may play different roles at different times during the same infection. Among the other temporally regulated genes in ZIKV infection, is the circadian clock gene PER2 that resembles HSPA5 (*Moni and Lio', 2017*).

## Validation of proviral and antiviral host factors

To probe the functional relevance of genes demonstrating correlations with DENV abundance, we first conducted loss-of-function screens. We measured the effects of siRNA-mediated depletion of 32 individual genes in Huh7 cells on DENV infection and on cellular viability (*Figure 5A* and *Figure 5—figure supplement 1A*). Using a cutoff of greater than 40% inhibition of viral infection as measured by luciferase assays normalized to cell viability in two independent screens, we identified multiple host factors that severely affect viral infection. These include a few components of the translocon previously shown to be essential for DENV: HM13 (*Marceau et al., 2016*) or WNV: SPCS2 (*Zhang et al., 2016*) as well as two novel components of the ER translocon: RPL31 and TRAM1 (*Ng et al., 2010*). Depletion of two proteins involved in membrane trafficking, TMED2 (secretory pathway) and COPE (retrograde, Golgi to ER) as well as the ER-resident chaperone and ERAD protein HSPA5 and the multifunctional transcription factor in ER stress, DDIT3 also reduced DENV infection.

In contrast, siRNA-mediated depletion of two genes that anticorrelate with intracellular virus abundance, ID2 and CTTNB1 (β-catenin), increased DENV infection, indicating that these proteins function as antiviral restriction factors, as previously reported in HIV (*Kumar et al., 2008*). Notably, ID2 and CTTNB1 are known interacting partners (*Rockman et al., 2001*), which may be acting via the interferon I pathway (*Hillesheim et al., 2014*). Suppression of another subset of overexpressed or underexpressed genes demonstrated no effect on DENV infection, suggesting that they were either non-essential or not restricting (possibly due to redundancy in host factors requirement) or that the level of knockdown was insufficient to trigger a phenotype.

To determine whether host factors found to be proviral are also rate limiting for infection, next we conducted gain-of-function screens. Huh7 cells ectopically expressing 30 of the 32 individual gene products were infected with DENV. Using a cutoff of greater than 30% increase in viral infection normalized to cell viability in two independent screens, we identified HSPA5, TMED2, SPCS2, and DDIT3 as factors whose overexpression increased DENV infection (*Figure 5B* and *Figure 5—figure supplement 1B*), indicating rate limitation associated with these important proviral factors. In contrast, overexpression of ID2 decreased DENV infection, indicating that ID2 has an antiviral

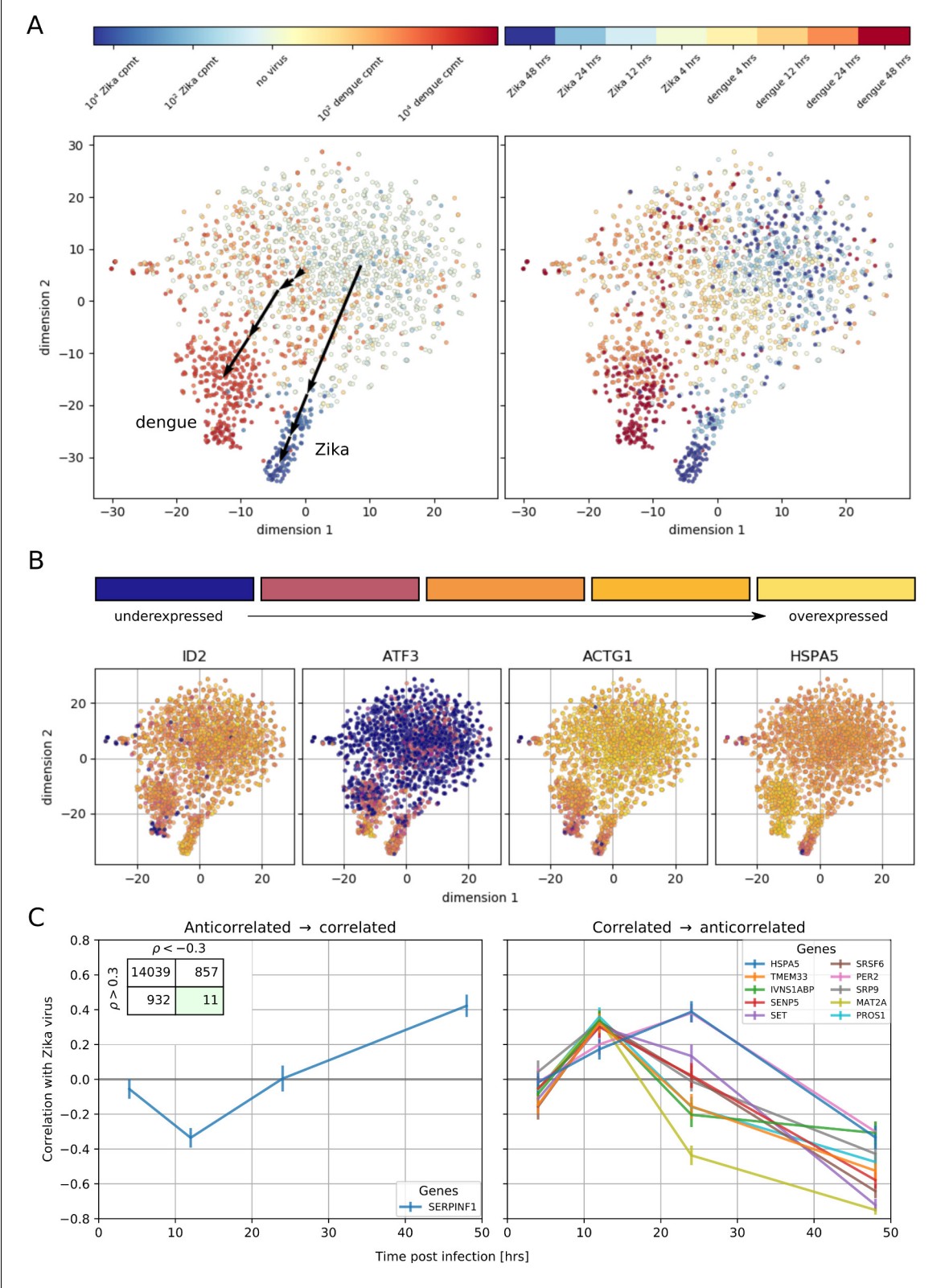

**Figure 4.** Temporally complex expression patterns during dengue and Zika infection. (**A**) t-SNE dimensionality reduction using all genes that correlate with at least one virus (<−0.4 or >0.4). Each dot is a cell and is colored by intracellular virus abundance (left panel) and time post-infection (right panel). Colors are shades of red for the dengue experiment, shades of blue for the Zika one. Arrows in the left panel indicate the average position of cells at increasing intracellular virus abundance. (**B**) Expression of four example genes as in *Figure 3B–E* on top of the t-SNE visualization. (**C**) Correlation

*Figure 4 continued on next page*

*Figure 4 continued*

between expression and Zika vRNA content switches from negative to positive (< −0.3 to >+0.3) for one gene (left panel) and in the opposite direction for 10 genes (right panel). Error bars are standard deviations of 100 bootstraps over cells. Unlike in dengue virus infection (**Figure 2G**), the temporal traces of Zika infection do not show a simple increase or decrease but rather complex dynamics.

DOI: https://doi.org/10.7554/eLife.32942.014

function. Overexpression of other proviral factors, such as COPE and TRAM1, decreased DENV infection, suggesting that DENV might be evolutionarily optimized for the natural expression level of these genes or that the observed correlation of these genes is not causative.

## Discussion

We have developed a new approach, designated viscRNA-Seq, to simultaneously quantify the whole transcriptome and intracellular virus abundance at the single cell level. This approach probes the quantitative gene expression dynamics of virus infections and is therefore complementary to knock-out and knockdown genetic screens, which induce a controlled perturbation (**Marceau et al., 2016**; **Zhang et al., 2016**; **Sessions et al., 2009**; **Kwon et al., 2014**; **Le Sommer et al., 2012**; **Lin et al., 2017**). However, unlike those loss-of-function assays, viscRNA-Seq is able to fully discern cell-to-cell variation within a single experimental condition, is compatible with time-resolved sampling, and can be used to study essential genes. Our approach can be easily adapted to any RNA virus, whether polyadenylated or not, by swapping a single oligonucleotide. Moreover, since RNA capture is highly efficient compared to droplet-based methods, an accurate quantification of both host gene expression and viral RNA (vRNA) can be obtained with as few as 400,000 sequencing reads per cell. Since full-length transcripts are recovered as in the original Smart-seq2 (**Picelli et al., 2014**) and unlike in droplet-based protocols, viscRNA-Seq can be combined with enrichment PCRs before sequencing to focus on specific host or viral factors at a fraction of the sequencing cost.

We have applied this high-throughput technique to study the temporal infection dynamics of DENV and ZIKV, two major global health threats (**Bhatt et al., 2013**). Our first finding is that beyond the expected increase in the number of infected cells in the culture over time, there is a large heterogeneity across cells from the same Petri dish. Since flavivirus replication is not synchronized, such heterogeneity might reflect host-responses at different stages of viral life cycle. The single-cell distributions of both intracellular virus abundance and gene expression indicate that mean values measured via bulk assays tend to over-represent highly infected cells. Moreover, bulk transcriptomics studies cannot account for uninfected cells and are therefore limited to high MOI (**Sessions et al., 2013**); in contrast, we are able to study both high-MOI and low-MOI cultures equally well and to separate the effect of MOI from the actual infection state of each cell.

We have leveraged the statistical power of sequencing thousands of cells to correlate intracellular virus abundance with gene expression across the whole human transcriptome. The genes with the strongest positive correlation with both viruses are members of the unfolded protein response (UPR), particularly the PERK branch, including DDIT3, ATF3, and TRIB3. The strongest negative correlates with both viruses are components of the actin and microtubule networks (e.g. ACTB, ACTG1, TUBB1) as well as members of nucleotide biosynthesis, suggesting a disruption of both cytoskeleton and cellular metabolism. The URP response starts abruptly once 1000 virus transcripts are present per million of total transcripts (i.e. when virus RNA comprises only 0.2% of the cellular mRNA); a threshold that is reached in most cells between 24 and 48 hr post-infection. Downregulation of cytoskeleton and metabolism, however, starts only at 20,000 virus transcripts per million of total transcripts; this higher threshold is reached in most cells at 48 hr post-infection. This delayed response may happen either because of direct cytopathic effects or as a consequence of the earlier UPR response, and is confirmed via parametric modelling (see Methods and **Figure 2—figure supplements 2,3**). Interestingly, a recent transcriptomics study also found ER stress pathways to be differentially regulated during DENV infection (**Sessions et al., 2013**). However, because thousands of host genes were classified as differentially expressed in that study, this overlap may be in part coincidental due to the sheer number of reported 'hits'. Indeed, the quantitative statistics resulting from the large number of single cell replicates was a key factor that enabled us to narrow down the list of potentially relevant genes to a small number that could be subsequently validated.

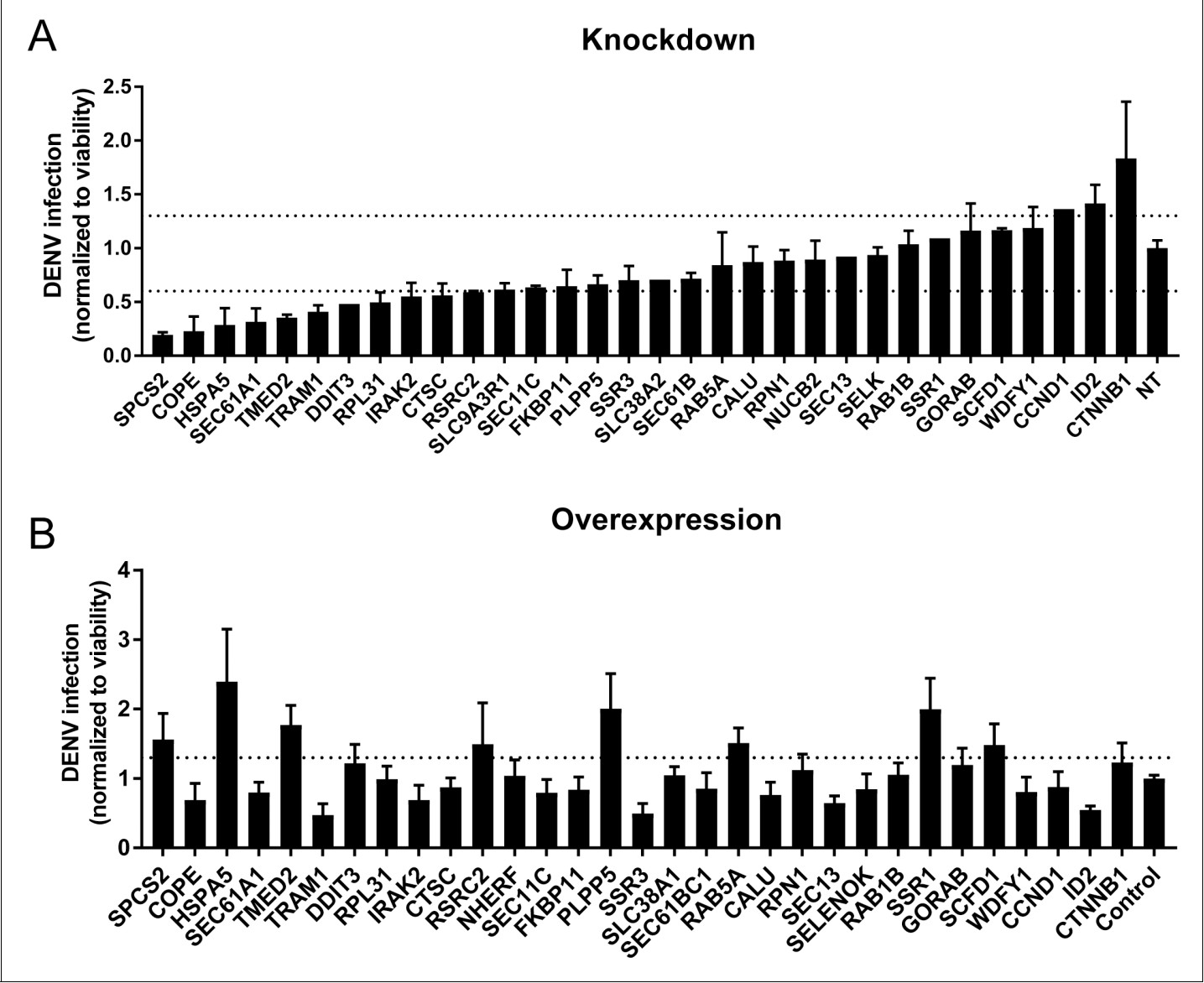

**Figure 5.** Validation of DENV proviral and antiviral candidate genes via siRNA-mediated knockdown and ectopic expression. DENV infection relative to NT siRNA (A) or empty plasmid (B) controls following siRNA-mediated knockdown (A) or overexpression (B) of the indicated host factors measured by luciferase assays at 48 hr post-infection of Huh7 cells and normalized to cell viability. Both data sets are pooled from two independent experiments with three replicates each. The dotted lines represent the cutoffs for positivity. Cellular viability measurements are shown in *Figure 2—figure supplement 2*.

DOI: https://doi.org/10.7554/eLife.32942.015

The following figure supplements are available for figure 5:

**Figure supplement 1.** siRNA (A) and ectopic expression (B) screens testing the involvement of the indicated host factors in DENV infection.
DOI: https://doi.org/10.7554/eLife.32942.016

**Figure supplement 2.** vRNA level versus gene expression across all time points and MOIs during DENV infection for 32 genes with interesting dynamics that were picked for validation via loss-of-function and gain-of-function experiments.
DOI: https://doi.org/10.7554/eLife.32942.017

It is noteworthy that at an MOI of 10, but not an MOI of 1, each cell is expected to be infected by more than one virus; however, we do not measure qualitative differences between the two MOIs except a faster and more robust increase of intracellular virus amount at the higher MOI. Moreover, although multiple rounds of infections are in theory possible with replication competent viruses, this

is expected to happen rarely since viruses from the Flaviviridiae family complete one replication cycle in at least 24 hr (*Ansarah-Sobrinho et al., 2008*; *Jones et al., 2005*; *Li et al., 2011Russell et al., 2008*); hence multiple rounds of infections are unlikely to be a significant factor in our analysis.

A number of host genes correlate strongly with one virus but correlate less or do not correlate with abundance of the other virus. Examples include subunits of several complexes involved in ER translocation and N-linked glycosylation: SEC61G, a subunit of the translocon; SSR3, a member of the TRAP complex; and OSTC, a subunit of the OST. Components of these three complexes were identified as essential host factors for DENV replication in a recent CRISPR-based knockout screen (*Marceau et al., 2016*). SEC11C, a subunit of the signal peptidase complex (SPCS), also behaves in this way, in agreement with the prior finding that this complex is essential for flavivirus infection (*Zhang et al., 2016*). Not all the subunits of these protein complexes correlate with virus abundance (*Figure 2—figure supplement 6*): for instance, whereas the catalytic OST subunits STT3A and STT3B show no correlation, other members such as MAGT1 show positive correlation in excess of 0.3, in agreement with recent findings (*Lin et al., 2017*). Strikingly, we do not observe a dominant enrichment of interferon-related genes among the most strongly upregulated during flavivirus infection (*Fink et al., 2007*). This result may be caused by virus-induced blocking of the interferon-induced signaling cascade (*Muñoz-Jordan et al., 2003*); moreover, Huh7 cells are known to activate the interferon cascade more mildly than other culture systems upon virus infection (*Guo et al., 2003*).

The expression of some host genes shows discordant correlation with DENV and ZIKV infection. Among the genes that are overexpressed during DENV infection but underexpressed during ZIKV infection are the molecular chaperone HSPA5 which has been shown to interact directly with the dengue E protein in liver cells (*Jindadamrongwech et al., 2004*), other members of the translocation machinery (SEC61B) or the TRAP complex (SSR1, SSR2), and ATF4, an ER-stress induced gene that interacts with DDIT3 and TRIB3. On the opposite end of the spectrum, genes that are underexpressed during DENV infection and overexpressed during ZIKV infection include the transcriptional regulator ID2. Both ID2 and cyclin D1 (CCND1), which is also strongly anticorrelated with DENV abundance, have been reported to be targets of β-catenin (*Rockman et al., 2001*; *Shtutman et al., 1999*).

Our analysis indicates that at large intracellular virus amounts the ER stress response is activated while cytoskeleton genes are underexpressed for both DENV and ZIKV; such profound expression changes could lead to apoptosis of infected cells. We attempted to incorporate dying cells as much as possible (see Methods). Cells with the largest intracellular DENV abundance show little change in the expression of apoptosis effector genes such as caspases, while their upstream regulators such as DDIT3 and TRIB3 are clearly overexpressed during late infection (see *Figure 3—figure supplement 1*), in line with a prior report (*Peña and Harris, 2011*). ZIKV seems to induce a similar response, with a few exceptions including upregulation of CASP3 (see *Figure 3—figure supplement 2*). From an evolutionary standpoint, keeping infected cells alive could benefit virus production. Alternatively, this lack of clear proapoptotic gene expression might be due to technical challenges in capturing and sequencing mRNA from dying cells or suggest regulation of ER-stress induced apoptosis at the protein level via posttranslational modifications (e.g. phosphorylation) rather than at the transcript level.

A few host genes (17 for DENV, 11 for ZIKV) show a complex dependence on time and intracellular virus abundance; at early time points, gene expression correlates positively (or negatively) with virus abundance, but this behavior is reversed at later time points. Among these genes are HM13, COPE, and SQSTM1 for DENV and HSPA5 for ZIKV. We speculate that these genes may play multiple roles during the virus replication cycle, acting as antiviral factors during certain phases of infection (e.g. cell entry) and as proviral during others (e.g. virion release). These genes may also represent virus triggered host-responses that were counteracted by viral proteins. Of these interesting hits, HM13 or signal peptide peptidase is involved in processing of signalling peptides after ER membrane translocation, a pathway that has been reported to be critical for several flaviviruses including dengue (*Zhang et al., 2016*). Furthermore SQSTM1, which is involved in selective autophagy, has been reported to affect DENV infection in a time-dependent manner, in agreement with our results (*Metz et al., 2015*), and to interact with the unrelated Chikungunya virus (*Judith et al., 2013*).

While viscRNA-Seq is a powerful tool to discover correlations between intracellular virus amount and gene expression, it does not directly address the underlying causal relations. A positive correlation between expression and virus abundance could represent either a preexisting higher expression level setting permissive conditions for infection or a consequence of the infection itself. Despite this limitation, it is possible to draw conclusion on a gene-by-gene basis: if the expression distribution in heavily infected cells shifts beyond the tails of the expression distribution in uninfected cells, it is likely that the expression change is a consequence of infection. This is exemplified by ACTB and DDIT3 (*Figure 2B–C*) and the positive correlation at late time points for COPE (*Figure 2F*). In other cases, for instance the negative correlation of COPE at early time points (*Figure 2F*), it is also possible that a stochastically lower expression of COPE was the cause and not the consequence of higher infection.

From the viscRNA-Seq screen we selected 32 candidate genes to determine whether they may play proviral or antiviral roles during DENV infection. The three genes HSPA5, SPCS2, and TMED2 showed clear proviral effects, reducing DENV replication upon knockdown and increasing it when overexpressed. The first two are known essential factors of DENV infection (*Jindadamrongwech et al., 2004*; *Zhang et al., 2016*), whereas TMED2, which is involved in coatomer complex (COPI) vesicle-mediated retrograde trafficking and trafficking from the golgi to the plasma membrane (*Fiedler et al., 1996*; *Goldberg, 2000*), has not been reported before. The gene ID2, which is an interaction partner of β-catenin, showed a strong antiviral effect, increasing DENV replication when knocked down and reducing it upon overexpression. Further in-depth studies are warranted to elucidate the role of the host factors TMED2 and ID2 in flavivirus infection. The hits SSR3, COPE, and TRAM1 reduced viral replication under both knockdown and ectopic expression, albeit at different degrees depending on the direction of the perturbation. This suggests that DENV might be evolutionarily adapted to wild type expression levels of these genes. Alternatively, the correlations for those genes may be not causative. Furthermore, it is striking that both TMED2 and COPE are involved in COPI-coated vesicle transport but produce opposite outcomes on DENV infection when overexpressed. Taken together with the time-switching correlation of COPE with intracellular DENV abundance, this result suggests a dual role for coatomer-coated vesicles during viral replication.

Overall, our study highlights the potential of single-cell level, high-throughput analyses to elucidate the interactions of human viruses with host cellular processes. Combining temporal information, cell-to-cell variability, cross-virus comparison and high-quality expression data has allowed us to identify pathways that react similarly to infection by dengue and Zika viruses, such as the unfolded protein response, and others that are more virus-specific. Furthermore, our findings reveal two proteins involved in ER translocation as novel host factors essential for DENV infection. Lastly, these results indicate that coatomer-coated vesicle trafficking shows both complex temporal behavior and includes a novel proviral factor, TMED2.

## Materials and methods

### Cells

Human hepatoma (Huh7) cells were obtained from Apath LLC (Brooklyn, NY). Cells were grown in DMEM (Mediatech, Manassas, VA) supplemented with 10% FBS (Omega Scientific, INC, Tarzana, CA), nonessential amino acid, 1% l-glutamine, and 1% penicillin-streptomycin (ThermoFisher Scientific, Waltham, MA) and maintained in a humidified incubator with 5% $CO_2$ at 37°C. C6/36 cells were obtained from ATCC (ATCC CRL-1660, Manassas, VA) and grown in Leibovitz's L-15 media (Mediatech, Manassas, VA) supplemented with 10% FBS (Omega Scientific, INC, Tarzana, CA, USA) and 1% HEPES (ThermoFisher Scientific, Waltham, MA) in a humidified chamber at 28°C and 0% $CO_2$. Cell lines identity was confirmed via phenotypic studies (low grade infection with hepatitis C virus (Huh7); syncytia formation upon DENV infection (C6/36) (*Corner and Ng, 1987*). Cells were tested negative for mycoplasma by the MycoAlert mycoplasma detection kit (Lonza, Morristown, NJ).

### Plasmids and virus constructs

The DENV 16681 infectious clone (pD2IC-30P-NBX) used in the single cell transcriptomic assays was a gift from Claire Huang (Centers for Disease Control and Prevention, Public Health Service, US

Department of Health and Human Services, Fort Collins, Colorado, USA)(*Huang et al., 2010*). A Renilla reporter DENV2 New Guinea C strain (NGC) plasmid (pACYC-DENV2) used in the validation assays was a gift from Pei-Yong Shi (University of Texas Medical Branch, Galveston, Texas, USA) (*Zou et al., 2011*). ZIKV PRVABC59 was obtained from BEI Resources. Open reading frames (ORFs) encoding 26 hits were selected from the Human ORFeome library of cDNA clones(*Rual et al., 2004*) (Open Biosystems), three from Addgene and one from DNASU(*Seiler et al., 2014*) and recombined into a pFLAG (for FLAG tagging) vector using Gateway technology (Invitrogen).

## Virus production

DENV2 16681 strain RNA was transcribed in vitro using mMessage/mMachine T7 kit (Ambion) from pD2IC-30P-NBX plasmid linearized by XbaI. DENV was produced by transfection of viral RNA into Huh7 cells and harvesting the culture supernatants at days 5–7. A Renilla reporter DENV2 NGC strain RNA was transcribed in vitro by mMessage/mMachine T7 kit (Ambion) from pACYC-Rluc2A-NGC linearized by XbaI. DENV was produced by electroporation of the viral RNA into BHK-21 cells and harvesting the supernatants at day 10. ZIKV, Puerto Rico strain (PRVABC59) was propagated in C6/36 insect cell. Titers of all viruses were measured via standard plaque assays on BHK-21 cells.

## Infection assays

Huh7 cells were infected with DENV or ZIKV for 4 hr at different MOIs (0, 1, and 10) and harvested at various time points post-infection. For the functional screens, Huh7 cells were infected with DENV in triplicates for 4 hr at MOI of 0.05. Overall infection was measured at 48 hr using standard luciferase assays.

## RNA interference

siRNAs (100 nM) were transfected into cells using silMPORTER (Millipore) 72 hr prior to infection with luciferase reporter DENV at MOI of 0.05. Custom Cherry-Pick ON-TARGETplus siRNA library against 32 genes was purchased from Dharmacon (see *Supplementary file 4* for gene and siRNA sequence details).

## Gain-of-function assays

Plasmids expressing ORFs encoding human genes or empty vector control were ectopically expressed in Huh7 cells by transfection with TransIT-LT1 (Mirus) 24 hr prior to infection with luciferase reporter DENV at MOI of 0.05.

## Viability assays

Viability was assessed using alamarBlue reagent (Invitrogen) according to the manufacturer's protocol. Fluorescence was detected at 560 nm on an Infinite M1000 plate reader (Tecan).

## Single cell sorting

At each time point, cells were trypsinized for 10 min, lifted them from the culture plate, pelleted and resuspended in 1 ml fresh media. After around 15 min, cells were pelleted again and resuspended in 2 ml 1X phosphate-buffered saline (PBS) buffer at a concentration of around 1 million cells per ml. Cells were filtered through a 40 um filter into a 5 ml FACS tube and sorted on a Sony SH800 sorter using forward and backscatter to distinguish living cells from dead cells and debris. Sorts were done into 384-well PCR plates containing 0.32–0.5 ul of lysis buffer (see below) using"Single cell' purity mode. A total of 12 384-well plates of single cells were sorted for the Dengue time course (four uninfected, 4 MOI 1, and 4 MOI 10), and eight plates for the Zika time course (4 uninfected and 4 MOI 1), yielding a total of about 7500 cells.

## Lysis buffer, reverse transcription, and PCR

To capture and amplify both mRNA and viral RNA (vRNA) from the same cell, the Smart-seq2 protocol was adapted (*Picelli et al., 2014*). All volumes were reduced by a factor 12 compared to the original protocol to enable high-throughput processing of 384-well plates. ERCC spike-in RNA was added at a concentration of 1:10 of the normal amount. The lysis buffer contained, in addition to the

oligo-dT primer at 100 nM final concentration, a virus specific reverse primer to capture the positive-stranded virus RNA at a concentration of 1 nM. The capture primer sequences were the following:

| Virus | Capture primer |
|-------|----------------|
| Dengue | AAGCAGTGGTATCAACGCAGAGTACGAACCTGTTGATTCAACAGC |
| Zika | AAGCAGTGGTATCAACGCAGAGTACTCCRCTCCCYCTYTGGTCTTG |

Different virus-specific primers and higher primer concentrations were tested but resulted in a large fraction of primer dimers. In order to reduce interference between the virus-specific primer and the Template Switching Oligo (TSO) used to extend the RT products, a 5'-blocked biotinylated TSO was used at the standard concentration. A large fraction of TSO concatemers was observed when testing reactions with a standard, non-biotinylated TSO. Reverse transcription (RT) and polymerase chain reaction (PCR) of the cDNA were performed in 1 ul and 2.5 ul, respectively: cells were amplified for 21 cycles. Lambda exonuclease was added to the PCR buffer at a final concentration of 0.0225 U/ul and the RT products were incubated at 37 C for 30 min before melting the RNA-DNA hybrid as it was observed that this reduced the amount of low-molecular weight bands from the PCR products. After PCR, the cDNA was diluted 1 to 7 in Tris buffer for a final volume of 17.5 ul. This dilution was used instead of the DNA purification by magnetic beads. In fact, we have tried to optimize purification by magnetic beads in 384-well plates but discovered that good libraries can be obtained without this step so we dropped it to maximize yield throughout the protocol, which allows fewer PCR cycles. All pipetting steps were performed using a TTPLabtech Mosquito HTS robotic platform.

## cDNA quantification

To quantify the amount of cDNA in each well after PCR, a commercial fluorimetric assay was used (ThermoFisher QuantIt Picogreen). Briefly, 80–300 nl of cDNA and 25 ul of 1:200 dye-buffer mix were pipetted together into a flat-bottom 384-well plate (Corning 3540). Six wells were used for a blank and five standard concentrations (0.1 to 2 ng/ul) in the same amount as the sample. The plate was briefly mixed, centrifuged, incubated in the dark for 5 min, and measured on a plate reader at wavelength 550 nm. cDna concentrations were calculated via an affine fit to the standard wells.

## Detection of infected cells by qPCR

Depending on the conditions (MOI and time since infection), the fraction of infected cells in each 384-well plate varies widely. In order to optimize sequencing on the widest possible dynamic range of virus amount per cell, we screen the amplified cDNA with a primer-probe based qPCR. Primer sequences are as follows:

| Virus | Forward primer | Reverse primer | Probe |
|-------|----------------|----------------|-------|
| Dengue | GARAGACCAGAGATCCTGCTGTCT | ACCATTCCATTTTCTGGCGTT | 6FAM-AGCATCATTCCAGGCAC-MGB |
| Zika | AARTACACATACCARAACAAAGTGGT | TCCRCTCCCYCTYTGGTCTTG | 6FAM-CTYAGACCAGCTGAAR-MGB |

The qPCR sequences for Dengue and Zika virus were adapted from (*Gurukumar et al., 2009*) and (*Faye et al., 2013*). For Zika, a minor groove binder (MGB) probe was used instead of the LNA probe of the original publication. Notice that for both viruses, conserved regions in the virus genome are selected and degenerate bases are used to ensure that the qPCR assay works independently on the mutations happening in the virus population during the cell culture. In addition to the virus-specific primers-probe, a commercial primer-probe assay for ACTB with a VIC fluorophore is used in the same reactions as an additional checkpoint for bona fide cDNA quality. 250 nl of each cell's cDNA were pipetted into a 5 ul reaction. The cycling protocol is 45 cycles of 95 C for 5 s followed by 60 C for 30 s. Synthetic single-stranded DNA sequences matching the qPCR primers-probe combinations were used in three concentrations (10 pM, 1 pM, 0.1 pM) and together with an additional blank well to calibrate the quantification (total of 4 wells for standards/blank). Each standard well included 250 nl of virus synthetic ssDNA and 250 nl of ACTB synthetic ssDNA covering the commercial assay, both at the same concentration. Notice that although RT-qPCR can be used directly

on cell lysates to obtain an accurate quantification of cellular RNAs, this assay is performed on pre-amplified cDNA instead, hence it is expected to be at best semi-quantitative. Nonetheless, we found it useful both as an early quality control step during the experiments and as a rough screening criterion to cherry pick cells for sequencing (see below). Because we obtained a great dynamic range of number of virus reads from single cells after sequencing, the qPCR results were not used in the downstream data analysis.

## Cherry picking of cDNA

Not all 7500 sorted cells were sequenced; rather, to improve coverage at the same cost, around 2000 cells were cherry picked for sequencing. With the results of the cDNA quantification and the virus and ACTB qPCR at hand, cells were selected such that they cover the largest possible set of conditions. For instance, in a plate with infected cells we ensured that both qPCR negative cells (ACTB but no virus), cells with little virus, and cells with a high amount of vRNA were all represented in the sequencing data. The selection was designed in a semi-automatic way via JavaScript and Python scripts and implemented on TTPLabtech Mosquito HTS and X1 HV robotic platforms. At the same time as cherry picking, the cDNA from each cell was also diluted to around 0.4 ng/ul for Tn5 endonuclease library prep. Although this concentration is slightly higher than usual or this type of libraries, the cDNA was not purified so that a certain fraction of the DNA is residual short oligos from previous reactions, which is most likely too short to end up on the sequencer.

## Library prep and sequencing

Sequencing libraries were prepared using the illumina Nextera XT kit following manufacturer's instructions, with the following exceptions: (1) we used a smaller reaction volume (around 1 ul per cell); (2) we chose a slightly higher cDNA concentration (0.4 ng/ul) as input, to compensate for the lack of bead purification upstream; (3) we designed, tested, and used a custom set of Nextera-compatible barcodes to increase plexity to 1536 cells per sequencing run, at an average depth of 250,000 reads per cell. The latter efforts allowed us to sequence each time course on a single illumina NextSeq sequencing run, reducing batch effects related to sequencing quality. We used the commercial 24 i7 barcodes and the 64 new i5 barcode sequences (see *Supplementary file 6*). We noticed a low level of cross-talk between these barcodes, indicated by up to five virus reads found in a few uninfected cells. However, considering that a sizeable fraction of cells in the same sequencing run (late infected and high MOI) had tens or even hundreds of thousand of virus reads, the amount of cross-talk between barcodes appears to be of the order of 1 in 10,000 or less. In terms of sequencing lengths, we sequenced eight bases from the standard i7 barcodes, 12 bases from the custom i5 barcodes, and 74 bases from each end of the insert (paired-end sequencing) using an illumina 150 cycles High Output kit for each of the two time courses.

## Bioinformatics pipeline

After sequencing was completed, we converted BCL files into gzipped FastQs via illumina's bcl2fastq. Because this software struggles with very high plexity libraries, we wrote a custom demultiplexer that copes better with the ~ 1000 cells per sequencing run of each time course. We then mapped the reads against the human GRCh38 genome with supplementary ERCC sequences using STAR Aligner (*Dobin et al., 2013*) and counted genes using htseq-count (*Anders et al., 2015*). Because the latter software was unmaintained at the time, one of us (FZ) took over the maintenance of the project, refactored the code, and added automated testing to check for software bugs. The reads that did not map to the human genome were remapped to the Dengue/Zika genome with rather permissive criteria using Stampy (*Lunter and Goodson, 2011*), filtered via custom scripts to eliminate artifacts, and counted to determine the viral reads per million transcripts (see below). The stanford high-performance computing clusters Sherlock and Sherlock 2.0 were used for the computations. Once the gene/virus counts were available, the downstream analysis was performed on a laptop using both custom Python scripts and the library singlet (https://github.com/iosonofabio/singlet; copy archived at https://github.com/elifesciences-publications/singlet), which is a second from-scratch implementation of the same functionality to minimize software bugs. The scientific data libraries numpy and scipy (*van der Walt et al., 2011*), pandas (*McKinney, 2011*), xarray (*Hoyer and Hamman, 2017*), SeqAn (*Döring et al., 2008*) and its derivative seqanpy (https://github.com/

iosonofabio/seqanpy) were used for number crunching. Matplotlib (*Hunter, 2007*) and seaborn (*Waskom et al., 2014*) were used for plotting. The gene expression and virus counts as well as the sample metadata are availble in *Supplementary file 7*. The virus particles and cell culture images in *Figure 1* are used under a Creative Common license from user Nossedotti and Y tambe at https://commons.wikimedia.org.

### Incorporation of dying cells

We attempted to incorporate dying cells as much as possible via the following experimental design choices: (i) we did not use a stain to distinguish between live and dead cells; (ii) the scattering gates used in the sorter enabled elimination of most debris particles, yet were kept as wide as possible, thereby enabling inclusion of dying cells; (iii) while cherry picking cells for sequencing, we intentionally kept cells with the largest virus/ACTB RNA ratio (as measured via the qPCR assays) to capture cells at late apoptotic stages.

### Error estimates and reproducibility

Correlation coefficients are computed as Spearman's rank correlation $\rho$. We estimate uncertainties by bootstrapping 100 times over cells and report the standard deviation in parentheses as errors on the last significant digit, or as error bars in graphs. To assess reproducibility, we performed an independent experiment on DENV infection on a smaller scale (1/5th of the cell numbers) and obtained consistent results (see *Figure 2—figure supplement 1*).

### Piecewise-linear fits of gene expression versus intracellular virus amounts

To quantitate the gene expression changes in response to virus infection, we fit a parametric model to the single cell values of gene expression versus intracellular virus amount, using the following equation:

$$log_{10}\, g = b + \Theta(v - v_t)\, (i + s \cdot log_{10} v),$$

where g is the expression of the focal gene in counts per million transcripts, v the intracellular virus amount in reads per million transcripts, $\Theta$ is the Heaviside step function that is zero for negative arguments and one for positive ones. The parameters are: b is the baseline gene expression level of uninfected cells, $v_t$ is the threshold, that is, the minimal intracellular virus amount required for gene expression to change, and i and s are the intercept and slope of the linear part of the curve, respectively. Minimization is performed via nonlinear least-squares. This model is arguably the simplest conceptualization of the thresholded response observed in out experiments for the genes with strongest correlation, see *Figure 2B–C* and *Figure 2—figure supplement 2*, and sheds light on the different thresholds for ER stress versus cytoskeleton gene sets, see *Figure 2—figure supplement 3*.

## Acknowledgements

This work was supported by award 1U19 AI10966201 from the National Institute of Allergy and Infectious Diseases (NIAID) to SE, 5T32AI007502 to EB, and grants from Stanford Bio-X and Stanford Institute for Immunity, Transplantation, and Infection. FZ is supported by a long-term EMBO fellowship (ALTF 269–2016). SYP was supported by the Child Health Research Institute, Lucile Packard Foundation for Children's Health, as well as the Stanford Clinical and Translational Science Award (CTSA, grant UL1 TR000093). We thank the anonymous reviewers for constructive comments.

## Additional information

### Funding

| Funder | Grant reference number | Author |
| --- | --- | --- |
| National Institute of Allergy and Infectious Diseases | 1U19 AI10966201 | Shirit Einav |
| Stanford Bio-X | | Shirit Einav |

| | | |
|---|---|---|
| Stanford Institute for Immunity, Transplantation, and Infection | | Shirit Einav |
| European Molecular Biology Organization | ALTF 269-2016 | Fabio Zanini |
| Child Health Research Institute | | Szu-Yuan Pu |
| Lucile Packard Foundation for Children's Health | | Szu-Yuan Pu |
| Stanford Clinical and Translational Science Award | UL1 TR000093 | Szu-Yuan Pu |
| National Institute of Allergy and Infectious Diseases | 5T32AI007502 | Elena Bekerman |

The funders had no role in study design, data collection and interpretation, or the decision to submit the work for publication.

## Author contributions

Fabio Zanini, Conceptualization, Data curation, Software, Formal analysis, Funding acquisition, Investigation, Visualization, Methodology, Writing—original draft, Project administration; Szu-Yuan Pu, Conceptualization, Funding acquisition, Validation, Investigation, Writing—review and editing; Elena Bekerman, Resources, Writing—review and editing; Shirit Einav, Conceptualization, Resources, Supervision, Funding acquisition, Validation, Investigation, Project administration, Writing—review and editing; Stephen R Quake, Conceptualization, Resources, Supervision, Funding acquisition, Investigation, Methodology, Project administration, Writing—review and editing

## Author ORCIDs

Fabio Zanini http://orcid.org/0000-0001-7097-8539
Shirit Einav http://orcid.org/0000-0001-6441-4171

## Decision letter and Author response

Decision letter https://doi.org/10.7554/eLife.32942.029
Author response https://doi.org/10.7554/eLife.32942.030

## Additional files

### Supplementary files

• Supplementary file 1. Number of cells processed and sequenced for each of the conditions - virus, time, MOI. Having around 100 high-quality cells within each experiment (2127 cells in total) allows for great statistical power compared to bulk assays which usually provide only a handful of replicates.
DOI: https://doi.org/10.7554/eLife.32942.018

• Supplementary file 2. Gene Ontology (GO) enrichment analysis for genes that are positively correlated (>=0.3) with intracellular virus abundance for both dengue and Zika virus highlights response to ER stress via the unfolded protein response (UPR), especially the PERK branch.
DOI: https://doi.org/10.7554/eLife.32942.019

• Supplementary file 3. Gene Ontology (GO) enrichment analysis for genes that are negatively correlated (<=−0.3) with intracellular virus abundance for both dengue and Zika virus indicates enrichment of metabolic processes including nucleotide biosynthesis and mitochondrial electron transport.
DOI: https://doi.org/10.7554/eLife.32942.020

• Supplementary file 4. Catalogue numbers, sequences, and other details on the siRNA probes used for the loss-of-function validation.
DOI: https://doi.org/10.7554/eLife.32942.021

• Supplementary file 5. ORFeome clones used for constructing overexpression plasmids. Genes without BC number means they are not available in Orfeome library and one of us (SYP) cloned their entries manually.
DOI: https://doi.org/10.7554/eLife.32942.022

• Supplementary file 6. i5 illumina-compatible index sequences for high plexity sequencing.
DOI: https://doi.org/10.7554/eLife.32942.023

• Supplementary file 7. Gene counts and metadata for all cells. This file is the recommended starting point for secondary analyses.
DOI: https://doi.org/10.7554/eLife.32942.024

• Transparent reporting form
DOI: https://doi.org/10.7554/eLife.32942.025

### Major datasets

The following dataset was generated:

| Author(s) | Year | Dataset title | Dataset URL | Database, license, and accessibility information |
|---|---|---|---|---|
| Fabio Zanini, Szu-Yuan Pu, Elena Bekerman, Shirit Einav, Stephen R Quake | 2018 | Single cell transcriptional dynamics of flavivirus infection | https://www.ncbi.nlm.nih.gov/geo/query/acc.cgi?acc=GSE110496 | Publicly available at the NCBI Gene Expression Omnibus (accession no. GSE110496) |

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
