## [Decision Letter]

Thank you for submitting your article "Single-cell transcriptional dynamics of flavivirus infection" for consideration by *eLife*. Your article has been reviewed by two peer reviewers, and the evaluation has been overseen by a Reviewing Editor and Arup Chakraborty as the Senior Editor. The reviewers have opted to remain anonymous.

The reviewers have discussed the reviews with one another and the Reviewing Editor has drafted this decision to help you prepare a revised submission.

Summary:

Your manuscript utilizes a novel method of single cell RNAseq (viscRNA-seq) to examine the relationship between the host transcriptome and the genome replication of two flaviviruses, DENV and ZIKV. This method generates individual RNAseq libraries from hundreds of infected cells and allows the relative quantification of both host transcripts and viral genomes within individual cells. You use this method to identify host transcripts that correlate with viral genome abundance at different times post-infection. This study identifies a handful of pro- and anti-viral factors, some of which had not been previously reported, and confirms many of these hits through targeted gene knockdowns and overexpression experiments. A comparison of DENV and ZIKV shows that the two viruses differ significantly in their specific patterns of host factor correlation, an intriguing and unexpected finding given how closely related these viruses are.

The main strength of this paper is that it details a novel and compelling approach for leveraging single cell RNAseq to dissect viral infection processes and host interactions in higher resolution than what is possible with bulk methods. As you point out, this method can be adapted to examine host factors that correlate with the replication of other virus families with only slight modifications to the method. Another strength is the comparison of DENV and ZIKV to define host factors that are common or unique to the two viruses.

The primary weakness of the study is that the interpretation of the results does not adequately account for the potential effects of cell death and multi-round replication. As a result, some of the conclusions need to be better tailored to suit the experimental results. Addressing the comments below is likely to ameliorate this weakness.

Essential revisions:

1) In interpreting their results, the authors do not appear to have accounted for cell death and secondary spread of the virus at late time points. Failing to synchronize infections and limit them to a single round means that there will be cells at many stages of the replicative cycle at late time points, making comparisons of viral genome content between individual cells difficult. This point is briefly noted in the discussion but warrants more serious consideration. Further, the death of infected cells at late time points may also skew results since only survivors will make it into the analysis.

2) The causality underlying the correlations is handled a little loosely. Do host transcripts correlate with viral genomes because cells with higher pre-existing expression of those factors are more permissive (and are thus "pro-viral"), or are those genes simply up-regulated in a dose dependent fashion by viral replication? This issue could also factor into the "time-switcher" phenotype.

3) Was the DENV experiment done more than once? The number of individual cells examined contributes an enormous amount of power within a single experiment, but it is important to define the amount of variation across experiments. Ideally, a correlation plot like Figure 3 could be shown for two independent DENV infections to demonstrate that correlations are reproducible for a given virus.

4) Another major comment that needs to be addressed is the lack of uncertainty estimates. The association between the viral RNA and relative gene expression levels is at the heart of the presented work, and it is not clear as to what fraction of the observed associations and trends can be attributed to chance. This impacts many of the presented results. Most notably:

a) Figure 2. Significance levels should be assessed based on the inter-plate variability (simply applying a Spearman rank correlation test will likely overestimate the significance). As significance levels will depend on the number of observations, an alternative representation of the figure may be needed – for instance, using a signed Z score on x axis (adjusted for multiple hypotheses).

b) Profile graphs in Figure 2 and Figure 4 need error bars. Some graphs may be too dense to show all the error bars, so the representation may need to be changed.

c) Relevant portions of the text where the numbers of associated genes are being assessed need to be suitably changed to reflect quantification.

d) A simpler confidence interval would also be useful for the individual rho estimates (e.g. Figure 2).

---

## [Author Response]

Essential revisions:1) In interpreting their results, the authors do not appear to have accounted for cell death and secondary spread of the virus at late time points. Failing to synchronize infections and limit them to a single round means that there will be cells at many stages of the replicative cycle at late time points, making comparisons of viral genome content between individual cells difficult. This point is briefly noted in the discussion but warrants more serious consideration. Further, the death of infected cells at late time points may also skew results since only survivors will make it into the analysis.

The virus strains used in this study are replication competent, hence indeed multiple rounds of replication are in theory possible. However, there is evidence that viruses from the *Flaviviridae* family, such as DENV and hepatitis C virus take at least 24 hours to complete one replication cycle (Ansarah-Sobrinho et al., 2008; Jones, Patkar, and Kuhn, 2005; Li et al., n.d.; Russell et al., 2008). Three out of four time points in our study are therefore from the first, single replication cycle, whereas the latest time point, at 48 hours post-infection, is a mixture of first and second replication cycles. Prior to initiating the time course experiments, all cells were split at the same time and infected with the viral stock synchronously. We have now discussed this in the revised manuscript, Discussion section.

We agree with the reviewers that cell death impacts population-level statistics of infected cells, because dead cells with no integral membrane sort and yield RNA at low efficiency. This skew is especially detrimental in bulk studies, whereas our single cell approach is less severely affected as long as one samples a (small) population of cells along the apoptotic path. By focusing on relatively early time points (up to 48 hours only), our experiment was designed to minimize death of DENV infected cells. Moreover, we attempted to incorporate dying cells as much as possible via the following experimental design choices: (i) we did not use a stain to distinguish between live and dead cells; (ii) the scattering gates used in the sorter enabled elimination of most debris particles, yet were kept as wide as possible, thereby enabling inclusion of dying cells; (iii) while cherry picking cells for sequencing, we intentionally kept cells with the largest virus/ACTB RNA ratio (as measured via the qPCR assays) to capture cells at late apoptotic stages. Nonetheless, we found little direct evidence of widespread apoptosis during DENV infection, even at MOI 10 and 48 hours post infection. Clear effectors of apoptosis (e.g. caspases) do not show a strong correlation with virus abundance, but DDIT3/CHOP and its downstream targets TRIB3 and PPP1R15A/GADD34, which are usually assumed as mediators of apoptosis, all show distinctive upregulation in cells with large intracellular virus content. DDIT3, TRIB3, and PPP1R15A have been proposed to be both anti- and proapoptotic genes, and their proapoptotic mechanisms of action remain elusive (Sano and Reed, 2013; Szegezdi et al., 2006). Moreover, many of the links in the signalling cascade induced by ER stress leading to apoptosis are regulated post-translationally (e.g. via phosphorylation), hence it is difficult to draw definitive conclusions from mRNA abundance beyond the remarkable absence of a consistent population of apoptotic cells in the face of multiple efforts to include it. Zika virus infection seems to induce a similar expression profile in terms of ER stress and apoptosis with the exception of CASP3, which is positively correlated with Zika content; ATF4, which is anticorrelated with Zika content; and EDEM1, which does not correlate with intracellular Zika virus amount. We have now added Figure 5—figure supplement 1 and Figure 5—figure supplement 2 and discussed this important point in the revised manuscript, see Discussion section.

2) The causality underlying the correlations is handled a little loosely. Do host transcripts correlate with viral genomes because cells with higher pre-existing expression of those factors are more permissive (and are thus "pro-viral"), or are those genes simply up-regulated in a dose dependent fashion by viral replication? This issue could also factor into the "time-switcher" phenotype.

We agree with the reviewers that viscRNA-Seq is designed to find correlations rather than direct causality. Nevertheless, we include data that favours more causal explanations for a limited set of genes. Overall, we report two distinct sets of genes in relation to intracellular virus abundance. The first set includes genes that show extreme correlations with dengue or Zika virus abundance and are exemplified by DDIT3 in Figure 2 and ACTB in Figure 2. For each of these genes, expression at high intracellular virus levels extends far beyond the expression distribution in uninfected cells, indicating that these changes are not just due to cells expressing preexisting “permissive” genes (see also the newly added Figure 3—figure supplement 1). The second set of genes are the temporal “switchers” for which either of the two scenarios or their combination is possible at different time points. For instance, the anticorrelated expression of COPE (Figure 2) at early time points lies roughly within the distribution of expression for uninfected cells, suggesting that cells may acquire less virus if they pre-express higher levels of COPE. In contrast, the positive correlation of COPE at 48 hours post-infection involves an expression beyond 1,000 per million transcripts, which is never observed in uninfected cells, supporting that this phenotype more likely represents a consequence of infection rather than a pre-existing expression status. Moreover, for a small subset of genes, we performed validation experiments in Figure 5 to investigate whether a depletion/overexpression of a highly correlated gene caused a change in infection level and found that some but not all correlated genes are causal in this sense. Future efforts to further understand the mechanism of the observed patterns should provide more insight on a gene-by-gene basis. This point is discussed in the revised manuscript, Discussion section.

3) Was the DENV experiment done more than once? The number of individual cells examined contributes an enormous amount of power within a single experiment, but it is important to define the amount of variation across experiments. Ideally, a correlation plot like Figure 3 could be shown for two independent DENV infections to demonstrate that correlations are reproducible for a given virus.

We have performed an independent, smaller scale (~ 1/5th of the cell numbers) DENV experiment and compared the results with those of the larger scale experiment presented in Figure 2. To address the reviewers’ comments, we have now included these results in Figure 2—figure supplement 6. Plotted are Spearman’s correlations between each gene and the number of virus reads for the small scale versus the largescale experiment (each dot is a gene). As indicated in the legend, for all genes that correlate above 0.3 in at least one experiment, the Pearson’s r between replicates is 0.82. Even when counting all expressed genes, a pool that generates a much larger noise level, the Pearson’s r between the independent experiments is above 0.65. The vast majority of the genes are thus behaving consistently in independent experiments. Only a few highly correlated genes fall outside the first and third quadrants: some of those are likely due to skewed sampling of the experimental conditions in the small-scale experiment. While this indicates that there is a small level of false positives – as one would expect – it is important to note that we rigorously validated functional relevance of genes downstream of our viscRNA-Seq method via knockdown and overexpression experiments. See Figure 2—figure supplement 6 and subsection “Incorporation of dying cells”.

4) Another major comment that needs to be addressed is the lack of uncertainty estimates. The association between the viral RNA and relative gene expression levels is at the heart of the presented work, and it is not clear as to what fraction of the observed associations and trends can be attributed to chance. This impacts many of the presented results. Most notably:a) Figure 2. Significance levels should be assessed based on the inter-plate variability (simply applying a Spearman rank correlation test will likely overestimate the significance). As significance levels will depend on the number of observations, an alternative representation of the figure may be needed – for instance, using a signed Z score on x axis (adjusted for multiple hypotheses).

We have estimated uncertainty in the values of correlation coefficient by bootstrapping cells 100 times and taking the standard deviation. Per the reviewers’ request, we now show it as an uncertainty on the last significant digit in panels Figure 2 and Figure 3. The standard deviation of the correlation coefficient for the typical gene is around 0.02-0.04, indicating that the top correlated genes (0.6) are 20 standard deviations away from zero, hence highly significant. The housekeeping gene PSMB2 in Figure 2, by comparison, is only around one standard deviation away from zero, so it should be considered uncorrelated (at our level of statistical power).

b) Profile graphs in Figure 2 and Figure 4 need error bars. Some graphs may be too dense to show all the error bars, so the representation may need to be changed.

We now have added error bars to Figure 2 and Figure 4. They are standard deviations of 100 bootstraps over cells and confirm that the time switching phenomenon is not due to technical noise.

c) Relevant portions of the text where the numbers of associated genes are being assessed need to be suitably changed to reflect quantification.

The bootstrapping analysis of uncertainty indicates that the error bars on correlation coefficients are mostly about 0.02, indicating that the top correlated (anticorrelated) genes yield robust estimates. We have included discussion of error bars in the main text, subsection “Error estimates and reproducibility”.

d) A simpler confidence interval would also be useful for the individual rho estimates (e.g. Figure 2).

We have now added uncertainty estimates to Figure 2 and the other panels via bootstrapping (see above for a detailed explanation).